# LLM-Prop: Predicting Physical And Electronic Properties Of Crystalline Solids From Their Text Descriptions

## Abstract

The prediction of crystal properties plays a crucial role in the crystal design process. Current methods for predicting crystal properties focus on modeling crystal structures using graph neural networks (GNNs). Although GNNs are powerful, accurately modeling the complex interactions between atoms and molecules within a crystal remains a challenge. Surprisingly, predicting crystal properties from crystal text descriptions is understudied, despite the rich information and expressiveness that text data offer. One of the main reasons is the lack of publicly available data for this task. In this paper, we develop and make public a benchmark dataset (TextEdge) that contains text descriptions of crystal structures with their properties. We then propose LLM-Prop, a method that leverages the general-purpose learning capabilities of large language models (LLMs) to predict physical and electronic properties of crystals from their text descriptions. LLM-Prop outperforms the current state-of-the-art GNN-based crystal property predictor by about 4% on predicting band gap, 3% on classifying whether the band gap is direct or indirect, and 66% on predicting unit cell volume. LLM-Prop also outperforms a finetuned MatBERT, a domain-specific pre-trained BERT model, despite having 3 times fewer parameters. Our empirical results may highlight the current inability of GNNs to capture information pertaining to space group symmetry and Wyckoff sites for accurate crystal property prediction.

## 1 Introduction

Predicting the properties of crystals is a problem with many useful applications, such as understanding the behavior and functionality of crystalline solids Tao et al. (2021); Parikh et al. (2022). Additionally, the ability to predict crystal properties would greatly accelerate the discovery and development of new crystals by identifying candidate materials warranting experimental study Meredig et al. (2014); Oliynyk et al. (2016); Raccuglia et al. (2016); Ward et al. (2016); Chen et al. (2019); Choudhary & DeCost (2021); Chen & Ong (2022).

Similarly to proteins Ioannidis et al. (2019); Strokach et al. (2020); Jiang et al. (2020); Shen et al. (2021); Wang et al. (2022); Jha et al. (2022); Réau et al. (2023), crystals are often represented as graphs that model the interactions between nearest neighbors; atoms in atomic crystals and molecules in molecular crystals Schütt et al. (2017); Xie & Grossman (2018); Chen et al. (2019); Choudhary & DeCost (2021); Chen & Ong (2022). For either case, crystal lattice sites are represented as nodes and the bonds (e.g., ionic or covalent or van der Waals) between them are represented as edges. Graph neural networks (GNNs) are then typically used to learn the contextual representation of each node and edge within the crystal graph to predict its properties. However, GNNs face several challenges to effectively predict crystal properties: 1) The need to efficiently encode the periodicity inherent to any crystal which results from the repetitive arrangement of unit cells within a lattice, a representation distinct from standard molecular graphs Yan et al. (2022). 2) The complexity of incorporating critical atomic and molecular information such as bond angles Choudhary & DeCost (2021) and crystal symmetry information such as space groups Kaba & Ravanbakhsh (2022). Finally, 3) graphs may lack expressiveness, which is useful in conveying complex and nuanced crystal information that is critical for accurate crystal property prediction.

The aforementioned challenges are due to the complexities of crystal graph representation. We aim to mitigate these challenges by modeling the crystal structure from its text description as opposed to its graph representation. Textual data contain rich information and are very expressive; additionally, incorporating critical/desired information in text is generally more straightforward compared to graphs. Crystal representations can for instance be learned by pretraining large language models (LLMs) on a large body of scientific literature which contains diverse chemical and structural information about crystal design principles and fundamental properties Beltagy et al. (2019); Walker et al. (2021); Huang & Cole (2022); Gupta et al. (2022). Then, one can finetune these pretrained LLMs on labeled data to solve specific tasks such as crystal property prediction Korolev & Protsenko (2023); Qu et al. (2023), crystal recommendation and ranking Qu et al. (2023), and synthesis action retrieval Song et al. (2023). However, these methods also face the challenges of limited pretraining and downstream data, limited computational resources, and a lack of efficient strategies to use the available resources.

In this work, we demonstrate effective strategies for the use of LLMs for the accurate prediction of crystal properties from their text descriptions. Existing approaches for finetuning LLMs for prediction tasks often either rely on both the encoder and the decoder or leverage encoder-only large models that tend to have as many parameters as encoder-decoder models. In this paper, we make the simple choice to use a pretrained encoder-decoder model, here T5 Raffel et al. (2020), entirely discard its decoder and finetune its encoder for regression and classification tasks. This has many desiderata: it allows us to cut the network size in half and importantly enables us to finetune on longer sequences and therefore account for longer-term dependencies in the crystal descriptions. With this simple yet effective choice, our approach outperforms not only large encoder-only models such as MatBERT, but also state-of-the-art GNN-based crystal property predictors such as ALIGNN.

**Contributions.** We make two main contributions, as summarized below:

- We collect, curate, and make public a benchmark dataset that contains approximately 144K crystal text descriptions and their properties.

- We propose LLM-Prop, an efficiently finetuned network that allows us to achieve state-of-the-art performance in crystal property prediction, outperforming the current best GNN-based crystal property predictors.

## 2 RELATED WORK

**GNNs for crystal property prediction.** One of the main GNN-based crystal property predictor is CGCNN Xie & Grossman (2018). CGCNN uses a convolutional neural network (CNN) on top of node embeddings from a crystal graph to learn the interactions between atoms in the crystal and predict crystal properties. Although CGCNN outperformed classical methods Jain et al. (2011); De Jong et al. (2015); Kirklin et al. (2015), it doesn't incorporate the many symmetries that crystals abide by. Several follow-up works aimed to incorporate those symmetries. Chen et al. (2019) proposed MEGNet, an architecture that incorporates crystal periodicity and only relies on GNNs to represent crystals. MEGNet outperforms CGCNN on various tasks; however, it still fails to account for critical information such as bond angles. ALIGNN Choudhary & DeCost (2021) was proposed to explicitly incorporate bond angles in addition to the other information accounted for by MEGNet and achieves state-of-the-art results compared to previous GNN approaches.

**Finetuned LLMs for crystal representation.** There have been very recent efforts using LLMs to learn representations of crystalline materials from their text descriptions. Qu et al. (2023) proposed an NLP-based crystal discovery framework that uses text embeddings as crystal representations. Crystal text descriptions were first fed into the in-domain pretrained LLMs to get embeddings. Then these embeddings were used to finetune a regression model for crystal property prediction. Finally, candidate crystals are recalled and ranked on the basis of the targeted properties. Several aforementioned in-domain pretrained LLMs were explored and MatBERT gave the best performance, a similar finding with Song et al. (2023). Even though these works rely on text to learn crystal representations, they mainly focus on ranking and recommendation. Property prediction was not thoroughly explored in these works; for example in Qu et al. (2023) the text embeddings were frozen and used only for regression when it comes to property prediction. Another work that is closely related to ours was proposed by Korolev & Protsenko (2023). Similarly to Qu et al. (2023), they finetuned MatBERT on text descriptions of crystals, although the focus was on classification tasks.

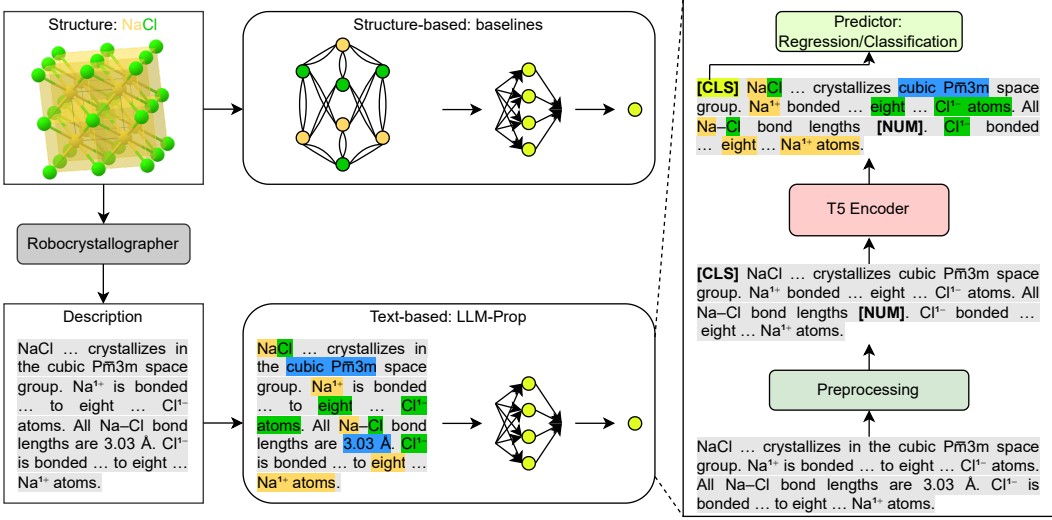

**Figure 1:** LLM-Prop architecture. On the left most of the figure, we show how we get the description from the crystal structure using Robocrystallographer. The middle part shows the comparison between our approach (text-based) and baselines (structure-based). The yellow colored information is related to Sodium (Na), green colored information is related to Chlorine (Cl), and blue colored information is related to additional information that text data provide such as space groups and bond distances. On the right most part of the figure, we then show our proposed LLM-Prop architecture (see 3.2 for more details).

Our work focuses on designing an efficient pipeline for the accurate prediction of crystal properties from text, for both regression and classification tasks. Our approach doesn't rely on in-domain pretrained LLMs, has significantly fewer parameters, and can even play a crucial role in designing a more robust ranking and recommendation system for crystals because getting text embeddings from a good property predictor can improve performance on those tasks. Furthermore, we perform extensive and carefully-designed experiments, comparing performance against the strongest GNN-based approaches to shed light on the benefits of using text and highlight the shortcomings of current GNN-based models for crystals. Finally, we release to the public the curated dataset we used as a benchmark, called *TextEdge*, to accelerate NLP for materials science research.

## 3 LLM-PROP

### 3.1 DATA COLLECTION AND ANALYSIS

We collected the dataset used in this work from the Materials Project database Jain et al. (2013) using the Materials Project free API[1] as of November 1, 2022. We focus on crystals with physical and electronic properties (band structures), which contain properties such as band gap, crystal volume, and an indicator of whether the band gap is direct or indirect. These properties have not yet been explored in other text-based methods and are usually hard to predict for GNN-based methods. The data contains $144,931$ crystals which we split into $125,098$ crystals for training, $9,945$ as a validation set, and $9,888$ as a test set. For each crystal, we collect its ID, structural information, band gap, volume, and whether its band gap is direct or indirect. We extracted the crystal text descriptions using Robocrystallographer Ganose & Jain (2019) (see Table 1 for examples).

### 3.2 LLM-PROP ARCHITECTURE

#### 3.2.1 T5

T5 Raffel et al. (2020) was the first unified language model that converts all text-based language problems into a text-to-text problem to perform both generative and predictive tasks. This framework is important for adapting and finetuning T5 on many tasks, and enables efficient multitask

---

[1]https://materialsproject.org/api

**Table 1:** Examples from the collected benchmark dataset.

| ID | Formula | Structure | Description | Band gap (eV) | Volume (Å³/cell) | Is-gap-direct? |
|---|---|---|---|---|---|---|
| mp-22851 | NaCl | | NaCl is Tetraauricupride structured and crystallizes in the cubic $P\bar{m}3m$ space group. $Na^{1+}$ is bonded in a body-centered cubic geometry to eight equivalent $Cl^{1-}$ atoms. All Na-Cl bond lengths are 3.03 Å. $Cl^{1-}$ is bonded in a body-centered cubic geometry to eight equivalent $Na^{1+}$ atoms. | 3.97 | 42.96 | No |
| mp-30274 | AcBrO | | AcOBr is Matlockite structured and crystallizes in the tetragonal P4/nmm space group. $Ac^{3+}$ is bonded in a 4-coordinate geometry to four equivalent $O^{2-}$ and five equivalent $Br^{1-}$ atoms. All Ac–O bond lengths are 2.49 Å. There are four shorter (3.40 Å) and one longer (3.54 Å) Ac–Br bond length. $O^{2-}$ is bonded to four equivalent $Ac^{3+}$ atoms to form a mixture of corner and edge-sharing $OAc_4$ tetrahedra. $Br^{1-}$ is bonded in a 5-coordinate geometry to five equivalent $Ac^{3+}$ atoms. | 4.24 | 140.14 | Yes |
| mp-570693 | $SbSe_3N_2Cl_7$ | | $SbCl_6N_2Se_3Cl$ is Indium-derived structured and crystallizes in the triclinic P1 space group. The structure is zero-dimensional and consists of two $N_2Se_3Cl$ clusters and two $SbCl_6$ clusters. In one of the $N_2Se_3Cl$ clusters, $N^{2-}$ is bonded in a bent 120 degrees geometry to one $Se^{2+}$ and one $Se^{3+}$ atom. The N–Se bond length is 1.75 Å. The N–Se bond length is 1.75 Å. $N^{3-}$ is bonded in a bent 120 degrees geometry to one $Se^{2+}$ and one $Se^{3+}$ atom. The N–Se bond length is 1.74 Å... | 1.70 | 787.76 | No |

finetuning. Raffel et al. (2020) carefully compared different architectures, pretraining objectives, unlabeled datasets, transfer approaches, and more on dozens of natural language tasks and then combined the best-performing approaches in each comparison to pretrain T5. For instance, while MatBERT, a BERT-based model, was pretrained using a masked language model (MLM) objective Devlin et al. (2018), T5 uses a span-masking objective Joshi et al. (2020) as it was shown to outperform MLM objective in terms of predictive power and speed. These considerations motivate our choice to use T5 as our main pretrained model.

### 3.2.2 ADAPTING T5 FOR PREDICTIVE TASKS

Transforming each task to text-to-text format requires T5 to use its decoder part to generate the output. While the decoder is necessary for generative and many downstream tasks, it adds unnecessary memory, time complexity, and work overhead when adapted for predictive tasks. Furthermore, Raffel et al. (2020) argued that the text-to-text format does not work well on regression tasks where the model is asked to generate the actual numerical value as the target instead of predicting the entire probability distribution. Raffel et al. (2020) resorted to only predicting the range to which a given value to be predicted belonged to instead of performing full regression withg T5.

How can we leverage T5 for highly accurate performance on predictive tasks, especially regression tasks? We propose LLM-Prop as an approach. LLM-Prop leverages T5 by directly adding a linear layer on top of its encoder for regression tasks. This linear layer can be composed with a sigmoid or softmax activation for classification tasks. Relying only on the T5 encoder reduces the total number of parameters by half which allows us to train on longer sequences and therefore incorporate more crystal information to improve predictive performance.

### 3.2.3 INPUT PROCESSING

**Removing stopwords.** We removed stopwords from the text descriptions as this preprocessing step has been shown to improve performance on predictive tasks Dieng et al. (2020); Niyongabo et al. (2020). We processed all publicly available English stopwords [2] and excluded them from the crystal text descriptions, except for digits and certain signs that may carry important information for crystal representation such as bond distances and angles.

**Replacing bond distance with a [NUM] token and bond angle with an [ANG] token.** Several works have shown that LLMs struggle to resolve contextual numerical information required for general common-sense reasoning Wallace et al. (2019); Zhang et al. (2020); Geva et al. (2020); Thawani et al. (2021). For example, Zhang et al. (2020) replaced numbers in the training data with their scientific notation using the [EXP] token (i.e., *314.1* with 3141[EXP]2) to train a new BERT model which outperforms the original BERT on tasks that require numerical reasoning skills such

---
[2]https://github.com/igorbrigadir/stopwords

as question answering. In our case, numbers not only add reasoning complexity but also increase the input sequence tokens since they are generally tokenized on a digit basis. We replace all bond distances and bond angles in the crystal text descriptions along with their units with [NUM] and [ANG] tokens, respectively. Those tokens are then added to the vocabulary as new tokens (i.e., we replace *3.03 Å* with [NUM] and *120 degrees or 120°* with [ANG]). While we are aware that this might also limit LLM-Prop to learn from the information related to bond distances and angles between atoms and molecules in the crystal structure, our results show that compressing the description as described—representing bond distances and angles with the same two special tokens described above—enables LLM-Prop to see more context in the text and achieve better performance.

**Prepending a [CLS] token to the input.** Devlin et al. (2018) showed that prepending a [CLS] token to every input, updating the embedding of that token together with the input tokens, and then using it for prediction improves predictive performance on downstream tasks. We used this same preprocessing step on T5, adding a [CLS] token in front of every input and also in the vocabulary, and then learning its representation jointly with the rest of the model. We used the embedding of the [CLS] token as the input to the linear layer for prediction.

### 3.2.4 LABEL SCALING

For regression tasks, we train LLM-Prop on normalized targets using different label normalization techniques: z-score normalization, min-max normalization and log normalization. Then we denormalize the outputs to calculate the prediction errors on the actual values. Suppose that we have $N$ crystals $X_1, \ldots, X_N$ and their corresponding properties $Y_1, \ldots, Y_N$ as a training set. Here $X_i$ denotes the text description of the $i^{\text{th}}$ crystal. Denote by $\mu$ and $\sigma$ the mean and standard deviation over all targets in the training set, respectively. Denote by $Y_{min}$ and $Y_{max}$ the minimum and the maximum target value in the training data, respectively. The three normalization techniques we explored normalize a given target value $Y_i$ as follows:

$$\hat{Y}_i(\text{z-score}) = \frac{Y_i - \mu}{\sigma}, \quad \hat{Y}_i(\text{min-max}) = \frac{Y_i - Y_{min}}{Y_{max} - Y_{min}}, \quad \text{and} \quad \hat{Y}_i(\text{log-norm}) = \log Y_i + 1.$$

## 4 EXPERIMENTS

We compare LLM-Prop with four baselines: three GNN-based models—CGCNN Xie & Grossman (2018), MEGNet Chen et al. (2019), and ALIGNN Choudhary & DeCost (2021)—and one text-based model, MatBERT Walker et al. (2021). The GNN-based methods are also trained and evaluated on the same benchmark dataset using the same splits to ensure a fair comparison. Note we do not include the results reported in the original articles since the dataset used was collected from an old version of the Materials Project, the 2018 version, and only contains about 70k crystals. We train all models using NVIDIA RTX A6000 GPUs. We next describe how we set up the experiments for all the baselines and for LLM-Prop.

### 4.1 BASELINES

**CGCNN.** We directly use the publicly available code[3] and the same experimental configurations as reported in the original paper to train a CGCNN from scratch and evaluate it on our benchmark dataset.

**MEGNet.** We train MEGNet from scratch and evaluate it on our benchmark dataset following the same implementation details and configurations described in the original article using the publicly available code released by the authors[4]. Unfortunately, for MEGNet, we could not find the implementation details on classification tasks, thus we only compared against it on regression tasks.

**ALIGNN.** Similarly to other baselines, we follow the original implementation details of ALIGNN released in the paper using the publicly available code from the authors[5] to train and evaluate it on our benchmark dataset.

**MatBERT.** We finetune MatBERT on our benchmark dataset using its original tokenizer. We initially preprocessed the input text descriptions of MatBERT as described in 3.2. However, these preprocessing steps led to worse performance for MatBERT. The possible reason might be that since

---

[3]https://github.com/txie-93/cgcnn

[4]https://github.com/materialsvirtuallab/megnet

[5]https://github.com/usnistgov/alignn

**Table 2:** Performance (MAE) comparison with the baselines on band gap prediction.

| 2*Model | 2*#Parameters | Band gap (eV) | |
| --- | --- | --- | --- |
| | | Validation set ↓ | Test set ↓ |
| **Structure-based models** | | | |
| CGCNN Xie & Grossman (2018) | - | 0.301 | 0.293 |
| MEGNet Chen et al. (2019) | - | 0.300 | 0.304 |
| ALIGNN Choudhary & DeCost (2021) | - | 0.249 | 0.250 |
| **Text-based models** | | | |
| MatBERT-zero-shot | 2*109.5M | 1.325 | 1.048 |
| MatBERT | | 0.244 | 0.249 |
| LLM-Prop-zero-shot (512 tokens) | 4*37M | 1.022 | 1.070 |
| LLM-Prop (512 tokens) | | 0.238 | 0.249 |
| LLM-Prop-zero-shot | | 1.117 | 1.031 |
| LLM-Prop | | **0.229** | **0.241** |

MatBERT can process only 512 tokens as input, the compressed information in 512 tokens that the preprocessing step gives, does not provide enough context for MatBERT. We therefore finetuned MatBERT with a batch size of 64 crystal descriptions, a learning rate of $5e-5$, and a dropout rate of 0.5 for 200 epochs using the Adam optimizer with a onecycle learning rate scheduler Smith & Topin (2019) for all the experiments.

### 4.2 LLM-PROP

For LLM-Prop, we finetune the original T5 encoder directly on our benchmark dataset without further pretraining it on domain-specific data. We first train the original T5 tokenizer on the benchmark data with the vocabulary size of 32k, then preprocess the data as described in 3.2. For regression tasks, we finetune LLM-Prop on normalized property values and denormalize the predicted values to calculate the error on the original values. Unless otherwise mentioned, we use z-score normalization when finetuning both LLM-Prop and MatBERT. Unless otherwise mentioned, we finetune LLM-Prop on 888 input tokens, using a batch size of 64, a learning rate of $1e-3$, and a dropout rate of 0.2 for 200 epochs using the Adam optimizer with a onecycle learning rate scheduler. For all models, we save the checkpoint of each epoch and evaluate on the test set with the checkpoint that gives the best performance on the validation set.

For regression tasks, we train both MatBERT and LLM-Prop with the mean absolute error (MAE) loss and evaluate them in terms of MAE error while the GNN-based models are trained with root mean square error (RMSE) loss and evaluated with the MAE error. For classification, we train them with binary cross entropy (BCE) loss and evaluate with the area under the ROC curve (AUC).

## 5 DISCUSSION

### 5.1 MAIN RESULTS

**LLM-Prop vs GNN-based models.** We find that LLM-Prop outperforms all GNN-based baselines on both regression and classification tasks. For band gap prediction, LLM-Prop outperforms the best-performing baseline (ALIGNN) on both validation and test sets by approximately $8\%$ and $4\%$ improvements, respectively (see Table 2). For volume prediction, LLM-Prop also improves over the best-performing baseline (ALIGNN) on both validation and test sets by approximately $67\%$ and $66\%$, respectively (see Table 3). The possible reason for this improvement might be that LLM-Prop can easily access the most important information for volume prediction, e.g. space group information, from the text descriptions compared to GNNs. This insight encourages two things: (1) the development of more text-based methods for crystal property prediction and the creation of high-quality text datasets needed for crystalline solids and (2) the development of new graph-based architectures that can incorporate more information, such as space group and Wyckoff sites. For predicting whether the band gap is direct or indirect, LLM-Prop also outperforms the best-performing baseline (ALIGNN) on the validation set with an approximate $5\%$ improvement and improves upon the best-performing baseline on the test set (CGCNN) by $3\%$ (see Table 5). Interestingly, finetuning

**Table 3:** Performance (MAE) comparison with the baselines on volume prediction.

| 2*Model | 2*#Parameters | Volume (Å³/cell) | |
| --- | --- | --- | --- |
| | | Validation set ↓ | Test set ↓ |
| **Structure-based models** | | | |
| CGCNN | - | 188.834 | 188.368 |
| MEGNet | - | 297.948 | 303.187 |
| ALIGNN | - | 129.580 | 126.486 |
| **Text-based models** | | | |
| MatBERT-zero-shot | 3*109.5M | 483.089 | 482.578 |
| MatBERT | | 49.761 | 53.282 |
| LLM-Prop-zero-shot (512 tokens) | 4*37M | 483.009 | 485.378 |
| LLM-Prop (512 tokens) | | 49.063 | 53.880 |
| LLM-Prop-zero-shot | | 482.863 | 485.396 |
| LLM-Prop | | **42.259** | **44.553** |

**Table 4:** Transfer learning performance comparison.

| 2*Model | Volume to Band gap (eV) | | Band gap to Volume(Å³/cell) | |
| --- | --- | --- | --- | --- |
| | Validation set ↓ | Test set ↓ | Validation set ↓ | Test set ↓ |
| **Structure-based models** | | | | |
| CGCNN-transfer | 0.300 | 0.295 | 187.473 | 182.997 |
| MEGNet-transfer | 1.461 | 1.472 | 190.013 | 195.664 |
| ALIGNN-transfer | 0.321 | 0.322 | 137.471 | 136.164 |
| **Text-based models** | | | | |
| MatBERT-zero-shot | 1.175 | 1.191 | 412.891 | 422.487 |
| MatBERT-transfer | 0.259 | 0.266 | 50.530 | 54.289 |
| LLM-Prop-zero-shot | 1.288 | 1.103 | 482.922 | 485.583 |
| LLM-Prop-transfer | **0.236** | **0.244** | **47.837** | **50.753** |

LLM-Prop with just 512 tokens as an input sequence length also outperforms all GNN-based baselines, despite not being able to account for enough context from the text descriptions (see Table 2, 3, and 5). Unfortunately, LLM-Prop is not able to get better performance when we do zero-shot predictions since T5 was not explicitly trained on in-domain data. However, the finetuned LLM-Prop (LLM-Prop-transfer) on band gap exhibits strong transfer learning ability compared to other models when transferred to volume prediction and vice-versa (see Table 4).

**LLM-Prop vs MatBERT.** For fair comparison, we first consider LLM-Prop and MatBERT when they are trained on the same sequence length (512 tokens). LLM-Prop yields better or comparable performance, including for zero-shot prediction, on all tasks (2, 3, and 5). When we finetune LLM-Prop on the maximum input sequence length it can process (888 tokens), it outperforms MatBERT by a large margin despite having 3× fewer hyperparameters. LLM-Prop improves upon MatBERT by approximately 6% and 3% on band gap prediction on the validation and test sets, respectively (see Table 2), 15% and 16% on volume prediction (see Table 3), and 19% and 18% on predicting whether the band gap is direct or indirect (see Table 5). Although MatBERT was pretrained on in-domain data, LLM-Prop shows stronger transfer learning ability when transferring between models finetuned on volume and band gap (see Table 4). The MatBERT results show that naively finetuning pretrained LLMs (even though they have been pretrained on domain specific data) is insufficient for accurate crystal property prediction and additional strategies such as those used in this paper are needed to achieve better performance.

**How much data does LLM-Prop need to achieve SOTA results?** We finetune LLM-Prop with randomly sampled data from the training set with the data size ranging from 5k to 90k and compare its performance with the baselines and the LLM-Prop model trained on the full data. The results in

**Table 5:** LLM-Prop performance (AUC ROC score) comparison with baselines on classifying whether the band gap value is direct or indirect.

| 2*Model | 2*#Parameters | Is-gap-direct | |
|---|---|---|---|
| | | Validation set ↑ | Test set ↑ |
| **Structure-based models** | | | |
| CGCNN | - | 0.797 | 0.830 |
| ALIGNN | - | 0.814 | 0.678 |
| | | | |
| **Text-based models** | | | |
| MatBERT-zero-shot | 3*109.5M | 0.500 | 0.535 |
| MatBERT | | 0.722 | 0.728 |
| LLM-Prop-zero-shot (512 tokens) | 4*37M | 0.494 | 0.484 |
| LLM-Prop (512 tokens) | | 0.839 | 0.836 |
| LLM-Prop-zero-shot | | 0.503 | 0.500 |
| LLM-Prop | | **0.856** | **0.857** |

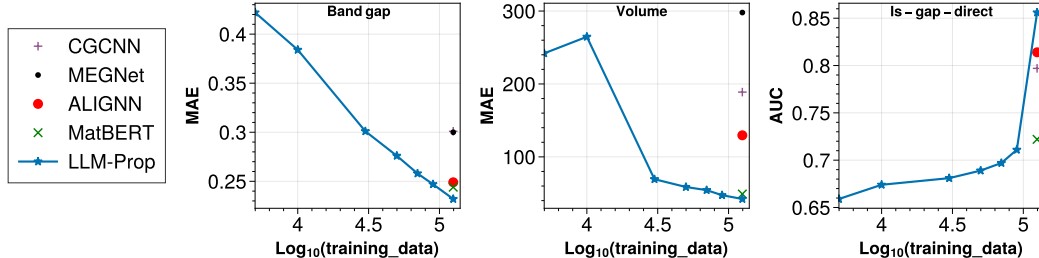

**Figure 2:** Training data size needed by LLM-Prop to achieve state-of-the-art (SOTA) results on predicting different properties. For instance, LLM-Prop achieves SOTA results on predicting band gap and volume with just about 90k training data ($log_{10}90k \approx 4.95$ **on the figure**) that corresponds to 35k fewer data than what baselines are trained on. We used log of the training data size on the x-axis for clarity. The performance of each model is calculated on validation set.

Figure 2 show that LLM-Prop can achieve SOTA results on predicting band gap and volume with just about 90k training data points ($log_{10}90k \approx 4.95$ on the figure) corresponding to 35k fewer data points compared to what baselines are trained on. Surprisingly, for volume prediction, LLM-Prop can outperform all GNN-based baselines with just 30k training data points ($log_{10}30k \approx 4.48$ on the figure) corresponding to about 95k fewer data points than what GNN-based baselines are trained on. These results highlight the efficiency and capabilities of LLM-Prop on predicting the properties of crystalline solids compared to the baselines.

## 5.2 ABLATION STUDIES

**How does each LLM-Prop design choice improve its performance?** Table 6 shows how each technique we use to make LLM-Prop a state-of-the-art crystal property predictor contributes to its performance. We first naively finetune a T5 encoder with its original tokenizer and without preprocessing the input (LLM-Prop-baseline) and then add to it each strategy separately. We also compare the performance of each strategy to when all strategies are combined together to make LLM-Prop (LLM-Prop+all). Overall, LLM-Prop+all provides the best results except on volume prediction where it shows comparable performance with the LLM-Prop+[CLS] version. Among all strategies, adding the [CLS] token for pooling seems to give the best improvement on all property prediction tasks. Replacing bond distances with the [NUM] token tends to slightly harm the performance, especially on volume prediction, compared to replacing bond angles with [ANG]. Label scaling significantly improves performance on both band gap and volume prediction while removing stopwords slightly improves the performance on band gap and volume prediction, but slightly harms the performance on predicting whether the band gap is direct or indirect. The modified tokenizer

**Table 6:** The contribution of each preprocessing strategy on LLM-Prop performance. We compare the baseline (when the input crystal descriptions and the targets are not touched, and with default T5 tokenizer) and to when all strategies are combined together.

| Model | Band gap ↓ | Volume ↓ | Is-gap-direct ↑ |
|---|---|---|---|
| LLM-Prop (baseline) | 0.256 | 69.352 | 0.796 |
| + modified tokenizer | 0.247 | 78.632 | 0.785 |
| + label scaling | 0.242 | 44.515 | N/A |
| + [CLS] token | 0.231 | **39.520** | 0.842 |
| + [NUM] token | 0.251 | 86.090 | 0.793 |
| + [ANG] token | 0.242 | 64.965 | 0.810 |
| - stopwords | 0.252 | 56.593 | 0.779 |
| LLM-Prop+all | **0.229** | 42.259 | **0.857** |

**Figure 3:** LLM-Prop performance per (a) label scaling strategy and (b) input sequence length. For the band gap and volume, the lower the better. While for the Is-gap-direct, the higher the better. The performance of each model is calculated on validation set.

slightly improves the performance of LLM-Prop on band gap prediction but harms the performance on predicting other properties.

**How does LLM-Prop perform with respect to the label scaling strategy?** We also analyze how each label scaling technique impacts performance on regression tasks. The results in Figure 3(a) show that when predicting band gap the performance difference among label normalization strategies is not significant while for volume prediction, z-score normalization (z_norm) significantly outperforms other normalization schemes.

**How does LLM-Prop perform with respect to the number of input tokens?** Figure 3(b) shows the performance of LLM-Prop as a function of the input sequence length. The results show that there is a clear correlation between performance and input sequence length. Accounting for longer sequences tends to improve performance. Therefore, by default, LLM-Prop is set to process up to 888 input tokens which is the maximum length that could be processed by the NVIDIA RTX A6000 GPUs that we used for training. And, as the results show, we believe that accounting for longer input sequence length will yield even better performance on crystal property prediction.

## 6 CONCLUSION

We introduced LLM-Prop, a carefully finetuned network derived from T5 for crystal property prediction. We showed, through an extensive set of experiments, that LLM-Prop achieves superior performance in predicting the physical and electronic properties of crystalline solids, outperforming the current state-of-the-art and ubiquitously used GNN-based architectures such as ALIGNN by a large margin on certain tasks. Our results highlight the great potential that text-based methods have in materials science and we release a benchmark text data, TextEdge, composed of text descriptions of crystals and their properties to encourage research in this nascent area.

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
