# OpenReview forum: "LLM-Prop: Predicting Physical And Electronic Properties of Crystalline Solids From Their Text Descriptions"
_ICLR.cc/2024/Conference — Submitted to ICLR 2024_

### Official Review · Reviewer_vjeb · 2023-10-30

**Soundness:** 3 good
**Presentation:** 3 good
**Contribution:** 2 fair
**Rating:** 5
**Confidence:** 4

**Summary:**

This paper proposed a novel LLM-based method for the prediction of crystal properties. The authors collect a dataset including crystal text descriptions with their properties, and use a T5-based finetune network to achieve SOTA performance on their benchmark. Also, the authors perform their model on two property prediction tasks with ablation.

**Strengths:**

(1). The authors conducted enough experiments to show the efficiency and capabilities of their model. The experiment is solid and convincing.
(2). This paper is well-written and easy to follow. The author’s motivation for this work is clear.
(3) The problem with crystal is important. And focusing on language and LLM is an important view for this problem.

**Weaknesses:**

1. Some details are not clear. How large is the pre-trained T5 model you used? The T5-small, T5-base, or T5-large? Specify this is important for the comparison of your efficiency.
2. The authors mentioned there’s some related work that also used finetuned LLMs for crystal representation. These works collect crystal text descriptions based on Robocrystallographer too. However, the authors did not compare their dataset with theirs about the text content and text quality. So, the experiment is not solid.

**Questions:**

(1). You have mentioned there’s some related work that also used finetuned LLMs for crystal representation. They collect crystal text descriptions based on Robocrystallographer too. Could you compare your dataset with theirs about the text content and text quality? Showing the advantage of your dataset is important for your contribution to data collection.

(2). Section 5.1 “The possible reason for this improvement might be that LLMProp can easily access the most important information for volume prediction, e.g. space group information, from the text descriptions compared to GNNs.” Could you please make some more ablation about your text information to prove your statement? Also, this is important to measure the importance of different types of information.

(3). Could you please make a task description to show the importance of the two tasks (Band gap and volume) in the experiments? Many other properties are predicted in related works such as energy per atom and bulk modulus. Why did you choose these two tasks?

---

> ### Author Response · Authors · 2023-11-16
>
> **Some details are not clear. How large is the pre-trained T5 model you used? The T5-small, T5-base, or T5-large? Specify this is important for the comparison of your efficiency.**
>
> > We did add the number of parameters for the T5 model and the MatBERT baseline on table 2 and 3, and table 5. We will specify that this corresponds to T5 small in the revision.
>
> **You have mentioned there’s some related work that also used finetuned LLMs for crystal representation. They collect crystal text descriptions based on Robocrystallographer too. Could you compare your dataset with theirs about the text content and text quality? Showing the advantage of your dataset is important for your contribution to data collection.**
>
> > Unfortunately, they did not release their data sets which makes it impossible to compare our benchmark with the data they used. Since we did not have access to their train-test splits, we finetuned MatBERT (the model that they finetuned on their data)  on our benchmark so as to make a fair comparison with LLM-Prop.
>
> **Section 5.1 “The possible reason for this improvement might be that LLMProp can easily access the most important information for volume prediction, e.g. space group information, from the text descriptions compared to GNNs.” Could you please make some more ablation about your text information to prove your statement? Also, this is important to measure the importance of different types of information.**
>
> > This is indeed a great point, we will include this ablation in the revision.
>
> **Could you please make a task description to show the importance of the two tasks (Band gap and volume) in the experiments? Many other properties are predicted in related works such as energy per atom and bulk modulus. Why did you choose these two tasks?**
>
> > Our motivation to focus on the physical properties (Band gap and Volume) was based on the fact that while these properties are very important they are understudied in many prior works probably because prior methods seem to not perform well on these two tasks.

---

> > ### Author Response · Authors · 2023-11-21
> > **Ablation results when removing the space group information from the input sequence to the LLM-Prop**
> >
> > Dear Reviewer *vjeb*, we thank you again for the constructive feedback on our work.
> >
> > We would like to share the performance results of LLM-Prop when it does not see the space group information to support the claim of why having the information in text descriptions that is not present in structure favors text-based property prediction methods (e.g LLM-Prop) over structure-based ones (GNNs). As the results show in the table below (the continuation of Table 6 in the paper), when we remove the space group information from the input sequence we see the performance drop on predicting all properties, with the highest degradation being seen on Volume prediction. We will also update the paper accordingly.
> >
> > | Model      | Band gap | Volume     | Is-gap-direct     |
> > | :---        |    :----:   |          :---: |           ---: |
> > | LLM-Prop (w/o space group info)      | 0.235       | 97.457  | 0.705|
> > | LLM-Prop   | **0.229**        | **42.259**| **0.857**|
> >
> > We hope that we addressed your questions and we would be happy to respond to any other further concerns.

---

### Official Review · Reviewer_Vdge · 2023-10-31

**Soundness:** 2 fair
**Presentation:** 3 good
**Contribution:** 1 poor
**Rating:** 3
**Confidence:** 3

**Summary:**

The paper studies the physical and electronic property prediction problem and proposes to use large language models to predict the properties of crystals from the text descriptions. The paper provides an input processing strategy to prepare the input text of the language model and adapt T5 for predictive tasks.

**Strengths:**

1. The studied problem is interesting and AI for science is also an important research area.
2. The paper is well-organized and easy to follow.

**Weaknesses:**

1. Although the task of crystal property prediction is an interesting problem for AI for science, the innovation of methodology is not surprising. This paper only uses the language model to make the prediction which is less novelty.
2. There are several recent works focusing on the text-rich graph where each node has text descriptions by jointly leveraging the GNN and LLM. These works are also related to the problem studied in this paper and should be discussed and compared.
3. The specific challenge in this problem is not clarified.

**Questions:**

1. What is the specific challenge in the problem of crystal property prediction?

---

> ### Author Response · Authors · 2023-11-16
>
> **Although the task of crystal property prediction is an interesting problem for AI for science, the innovation of methodology is not surprising. This paper only uses the language model to make the prediction which is less novelty.**
>
> > Although the idea of using LMs to make predictions is not novel, finetuning LLMs to achieve SOTA results on crystal property prediction using their descriptions compared to domain-specific models requires innovative approaches and to the best of our knowledge, LLM-Prop is the first method that is based on finetuning a non-domain-specific LLM to achieve that. Different techniques (see Table 6) that we combine together to finetune T5 encoder enable LLM-Prop to outperform a naively finetuned MatBERT despite having a three times fewer parameters than MatBERT which has been pretrained on domain-specific data. As the results show, we also improved over the GNN-based methods, achieving a significant difference in performance (see Table 3-5).
>
> **There are several recent works focusing on the text-rich graph where each node has text descriptions by jointly leveraging the GNN and LLM. These works are also related to the problem studied in this paper and should be discussed and compared.**
>
> > Although there are works that use GNN and LLM, for example [Zhao et al. 2023](https://openreview.net/forum?id=q0nmYciuuZN), we would like to remind the reviewer that these works are specifically implemented for tasks that are related to text-attributed graphs (for e.g. knowledge graphs and paper citation graphs) such as node classification. In those works each node is a text/document and nodes are interconnected in a graph format. Our work specifically deals with the task of crystal property prediction where each crystal is represented by a text description and does not require building a knowledge graph between different crystals, it directly predicts crystal properties using a finetuned LLM on a text description of that crystal. Although simple, our approach achieves state-of-the-art performance, outperforming even state-of-the-art GNN-based methods.
>
> **The specific challenge in this problem is not clarified. What is the specific challenge in the problem of crystal property prediction?**
>
> > Crystals, unlike molecules, are rigid bodies. They have strong symmetries that are challenging to capture by even the current state-of-the-art GNNs ([Choudhary et al. 2021](https://www.nature.com/articles/s41524-021-00650-1)). Our work shows that working in the description space of these crystal structures is a promising way to predict the properties of crystals. The text descriptions that are the input to our proposed method already have information about symmetries such as space groups and by having a powerful pretrained language model like T5 that can easily understand text and extract useful information from the crystal descriptions, we get the crystal representation that is used to accurately predict its properties.

---

> > ### Author Response · Authors · 2023-11-21
> >
> > Dear Reviewer *Vdge*, we appreciate your constructive feedback on our work.
> >
> > We would like to check if we were able to address your concerns.

---

### Official Review · Reviewer_WsXq · 2023-10-31

**Soundness:** 2 fair
**Presentation:** 2 fair
**Contribution:** 2 fair
**Rating:** 3
**Confidence:** 3

**Summary:**

This paper investigates the task of predicting crystal properties with LLM decoder-only model. The model is trained on the constructed benchmark dataset (TextEdge) which contains crystal structure, description, properties such as band gap. The experimental result shows that the proposed method LLM-Prop is slightly better than previous method MatBERT and ALIGNN.

**Strengths:**

- The paper extends the previous crystal representations task based on text into the property prediction task based on text.
- The benchmark dataset TextEdge should be helpful in the predicting crystal properties domain.

**Weaknesses:**

- My top concern is the technical depth of the paper. For the method, this paper is an implementation for predicting crystal properties task with decoder part of T5. The discussion of choice of T5 is weak and Input Processing is also trivial.
- For the Data collection in Sec 3.1, I did not see the challenging part and discussion about the process of data collection including quality control. I am not sure about the quality of data at all.

**Questions:**

N/A

---

> ### Author Response · Authors · 2023-11-16
>
> **My top concern is the technical depth of the paper. For the method, this paper is an implementation for predicting crystal properties tasks with the decoder part of T5. The discussion of choice of T5 is weak and Input Processing is also trivial.**
>
> > We did not use the decoder part of T5 but rather built on the encoder of T5 with a carefully designed finetuning approach. We refer the reviewer to the last paragraph of the Introduction section and section 3.2.1 and 3.2.2  for our detailed discussion of why we chose T5. We also refer the reviewer to section 5.2 and Table 6 that discuss the detailed contribution of the input processing to the overall performance of LLM-Prop. Please, note that the input processing is very crucial when handling scientific data. For instance this very recent work ([Golkar et al. 2023](https://arxiv.org/abs/2310.02989)) shows that encoding numbers in scientific text differently improves the LLMs performance substantially.
>
>
> **For the Data collection in Sec 3.1, I did not see the challenging part and discussion about the process of data collection including quality control. I am not sure about the quality of data at all.**
>
> >This is a good point. We will indeed add more details for this part in the revision. We will add details on how Robocrystallographer works, how long it took us to collect the structure data from the Materials Project database, and how to generate the descriptions with Robocrystallographer. We also add details on quality control where for instance we discarded crystals with shorter descriptions (less than 5 tokens), and statistics of the final datasets such as the average length of the descriptions, total number of tokens, average number of numerical values in each description, etc.

---

> > ### Author Response · Authors · 2023-11-21
> >
> > Dear Reviewer *WsXq*, thanks again for your constructive feedback on our work.
> >
> > We would like to check if we were able to address your concerns.

---

### Author Response · Authors · 2023-11-16
**General response**

- We thank the reviewers for their comments and questions.
- Generally, we think there may be a misunderstanding of where this work lies in the current AI literature. While we are not proposing a new architecture, which would be a goal if the paper were purely for an NLP or AI audience, we are bringing new results and insights into the literature on AI for science. More specifically, we are proposing an approach to crystal property prediction, a very important task in materials science,  based on text descriptions and showing state-of-the-art results compared to graph neural network approaches which are the current standard approach to property prediction. Our work brings the tools of NLP to the world of materials science and showcases the importance of text as data for approaching property prediction problems.
- Our key contributions are as follows:

     - Creating a benchmark dataset (and making it publicly available) that can be used to advance the research of  NLP applications in materials science.
     - Proposing an efficiently finetuned language model that allows us to achieve state-of-the-art performance in crystal property prediction.
     - Analyzing how different NLP techniques contribute to the performance of finetuned language models for crystal property prediction.
     - Our experimental results highlight that a carefully finetuned non-domain-specific pretrained language model can outperform a naively finetuned domain-specific that is 3x bigger in size. Our results challenge the current common practice in AI for science consisting in pretraining domain-specific models, which is costly.
- We provide a response to all the questions and comments by reviewers below.

---

### Meta-Review · Area_Chair_n4A3 · 2023-12-07

**Metareview:**

This paper presents a benchmark dataset (TextEdge) and a method called LLM-Prop that leverages large language models to predict physical and electronic properties of crystals from their text descriptions, outperforming state-of-the-art GNN-based crystal property predictors and domain-specific pre-trained BERT models. However, the consensus among reviewers is that the paper is not ready for acceptance in its current form. The methodolgy may seem to be incremental (it seems to be a relatively straight-forward approach to extend the current AI models into a new application area). I recommend not to accept the paper at this stage.

**Justification For Why Not Higher Score:**

n/a

**Justification For Why Not Lower Score:**

n/a

---

### Decision · Program_Chairs · 2024-01-16

Reject